# The Proteasome Activator PA200/PSME4: An Emerging New Player in Health and Disease

**DOI:** 10.3390/biom12081150

**Published:** 2022-08-20

**Authors:** Ayse Seda Yazgili, Frédéric Ebstein, Silke Meiners

**Affiliations:** 1Comprehensive Pneumology Center (CPC), Helmholtz Center Munich, Max-Lebsche Platz 31, 81377 Munich, Germany; 2Institut für Medizinische Biochemie und Molekularbiologie (IMBM), Universitätsmedizin Greifswald, Ferdinand-Sauerbruch-Straße, Klinikum DZ/7, 17475 Greifswald, Germany; 3Research Center Borstel/Leibniz Lung Center, Parkallee 1-40, 23845 Borstel, Germany; 4Airway Research Center North (ARCN), German Center for Lung Research (DZL), 23845 Sülfeld, Germany; 5Institute of Experimental Medicine, Christian-Albrechts University Kiel, 24118 Kiel, Germany

**Keywords:** PA200, PSME4, proteasome

## Abstract

Proteasomes comprise a family of proteasomal complexes essential for maintaining protein homeostasis. Accordingly, proteasomes represent promising therapeutic targets in multiple human diseases. Several proteasome inhibitors are approved for treating hematological cancers. However, their side effects impede their efficacy and broader therapeutic applications. Therefore, understanding the biology of the different proteasome complexes present in the cell is crucial for developing tailor-made inhibitors against specific proteasome complexes. Here, we will discuss the structure, biology, and function of the alternative Proteasome Activator 200 (PA200), also known as PSME4, and summarize the current evidence for its dysregulation in different human diseases. We hereby aim to stimulate research on this enigmatic proteasome regulator that has the potential to serve as a therapeutic target in cancer.

## 1. Introduction

The proteasome activator 200 (PA200: protein name, or PSME4: gene name) is a 200 kDa large monomeric protein that binds to the 20S and 26S proteasome complexes and activates its proteolytic activities towards peptides [1]. It is highly conserved amongst mammals, and its homologs are present in worms (*Caenorhabditis elegans*), plants (*Arabidopsis thaliana*), and yeast (*Saccharomyces cerevisiae*), but not in *Drosophila melanogaster* [2]. However, the sequence similarity between human PA200 and yeast PA200, named Blm10, is only 17% questioning the functional similarity, despite the structural conservation [3,4]. The analysis of PA200/PSME4 has been hampered by the lack of specific antibodies [5,6], and its GC-rich DNA sequence is impeding cDNA cloning and overexpression. Moreover, the mice lacking PA200/PSME4 do not display a prominent phenotype except for male infertility [7,8]. This dampened the scientific enthusiasm for this proteasome activator upon its discovery twenty years ago. The recently published cryo-EM structures of human PA200 [4,9], its proposed role in the degradation of acetylated histones [10,11], and the reports on PA200/PSME4 dysregulation in disease [5,6,12] have, however, spiked novel interest in understanding the cellular function of PA200/PSME4 and its potential as a therapeutic target in hyperproliferative diseases. The present review will critically discuss the available evidence on mammalian PA200/PSME4, its function, and its dysregulation in disease with the aim of stimulating research on this enigmatic proteasome regulator.

## 2. Expression and Regulation of PA200

PA200/PSME4 contains a putative nuclear localization signal that indicates its nuclear localization [1]. The cross-linking experiments, using cytoplasmic, microsomal, and nuclear extracts, detected PA200 in all three subcellular fractions [13]. Its subcellular localization may thus vary depending on the cell type and cellular function. Several studies observed the colocalization of PA200 with genomic DNA [10,14]. The CHIP seq analysis of the SH-SY5Y neuroblastoma cells found PA200 to be associated with the transcription start site of multiple gene promoters [15].

According to the consensus dataset of the human protein atlas (HPA), the RNA expression of PA200 is highest in the tongue, skeletal muscle, and testes [6]. In contrast, the expression in immune cells is generally low. On the protein level, the data are less reliable. The antibodies used for cell or tissue staining or Western blot analyses are of limited specificity, as suggested by the HPA validation assays [6]. Indeed, Welk et al. recently discovered that the leading commercially available antibody used in most of the studies is unspecific in Western blot and immunohistochemistry analysis (Figure 1A–C; Antibody #1). They dissected the specificity of the commercially available anti-PA200 antibodies, using either PA200 gene silencing in human cells (Figure 1A,D) or testis tissue from PA200 knockout (KO) mice (Figure 1B,C,E,F) [5]. In PA200-deficient mice, the PA200 gene was depleted except for the first coding exon, which could potentially encode the N-terminal 80 amino acids of the 1,869-amino-acid long PA200 protein [7]. Other commercially available antibodies (#2–4) proved, however, to be specific in the detection of the 200 kDa large PA200 protein.

This specificity analysis also challenges the early findings by Ustrell and Blickwedehl that indicated the existence of multiple isoforms of bovine and human PA200 isolated by cDNA cloning and Western blot analysis, using the above-described antibody #1 [1,14]. In accordance with Welk’s data, the latest entry from the ensemble database denotes only two protein isoforms for the human PA200/PSME4, i.e., the full-length protein and a short version of the 571 amino acids predicted to undergo nonsense-mediated decay [16].

Not much is known about the transcriptional regulation of PA200 expression. The data from Sha et al. reported the transcriptional induction of PSME4 as part of an autoregulatory feedback loop upon proteasome inhibitor treatment [17,18]. This was confirmed by Welk et al., 2016, demonstrating that PSME4 was upregulated two-fold within 24 h after the inhibition of the proteasome with bortezomib, or upon impaired 26S proteasome assembly after the silencing of the 19S subunit Rpn6 [19]. The in silico analysis of the PSME4 promotor confirmed the presence of a highly conserved Nrf1-binding site close to the PSME4 transcriptional start site [19]. We also demonstrated the transcriptional activation of PA200 in response to transforming growth factor beta (TGF-β) [5]. The same study showed the downregulation of PA200 upon the differentiation of the basal stem cells into differentiated airway epithelial cells. The data mining for the transcriptional regulation of PA200, using the Expression Atlas or other tools, is recommended to explore the evidence for transcriptional regulation in transcriptomics datasets. Regarding the post-transcriptional regulation of PA200, a single study reported the binding of the microRNA-29b at the 3′ UTR of PA200 [20], but confirmatory data are missing. On the proteasome complex level, the assembly of PA200 into the 20S or 26S proteasome complexes occurs rapidly, as demonstrated in response to acute proteasome inhibition. These data suggest the existence of free PA200 that can be rapidly recruited to the 20S and 26S proteasome to form singly or doubly-capped PA200/20S or hybrid PA200/26S complexes [19]. PA200 was also previously reported to form PA200-20S-19S proteasome complexes upon irradiation in HeLa cells [21]. A recent BioRxiv paper also captured the structures of PA200-20S-PA28, as well as the PA200-20S-19S complexes [22]. The cross-linking proteomics analysis from the lab of Marie-Pierre Bousquet confirmed the presence of PA200 as part of the 20S, but also of the 26S complexes. According to their cross-linking data, PA200 locates in the cytoplasm and the nucleus at a similar ratio [13]. It makes up under 5% of the entire proteasome fraction in the cell [23]. A new BioRxiv manuscript by the Merbl lab indicated that PA200 could also bind to 20S complexes containing immunoproteasome subunits [12]. The authors suggest that the binding of PA200 to the immunoproteasome might counter-regulate the immunoproteasome-specific activities involved in the MHC class I antigenic peptide generation.

## 3. Structure

The first PA200/20S structure was revealed in 2005, with PA200 isolated from the bovine testes [24]. Three different particles were identified with 23Å resolution: 20S alone; 20S-singly capped with PA200; and 20S-doubly capped with PA200 at a ratio of 50:40:10. PA200 was described as an asymmetric, hollow, dome-like structure bound to all the outer alpha subunits of the 20S (except α7) as a monomer, thereby opening the 20S gate. A recent structure of human PA200 was released in late 2019 in combination with fully in vitro reconstituted human 20S [4]. In this study, all α and β subunits of the 20S were expressed using a baculovirus expression system together with five proteasome assembly chaperones that facilitated the in vitro assembly of the 20S. A third baculovirus was used to co-express the human PA200. The high-resolution cryo-EM analysis of this recombinant PA200/20S complex showed a similar dome-like structured PA200 as before, that forms by helical repeats and binds to the α-subunits of the 20S proteasome (Figure 2A). The interaction of PA200 with the alpha subunits was resolved in high detail: PA200 bound to 20S via two anchor points: one close to the α5-α6 interface and the other at the α1-α2 interface. Upon the binding of PA200, α5-α7 relocated to the inner surface of the PA200 dome, whereas α3 relocated to have a wider α ring-opening upon PA200 binding. These changes in the α-subunits resulted in allosteric effects on the catalytically active β-subunits, with the β2 active site widening, while the β1 and β5 sites narrowed. Following these structural changes of the catalytic centers, the binding of PA200 to the recombinantly expressed 20S resulted in increased trypsin-like (T-L) (β2) and decreased chymotrypsin-like (CT-L) (β5) and caspase-like (C-L) activities (β1) in vitro. The latest cryo-EM structure of the PA200 (3.75 Å) and PA200-20S (2.72 Å) complex was released in 2020 [9]. In this study, the recombinantly expressed human PA200 was complexed with commercially available 20S standard proteasomes and yielded a heterogeneous mix of doubly- or singly capped PA200/20S complexes. The 20S α-subunits were similarly re-arranged upon PA200 binding, as described by Toste-Rego, while no rearrangement was observed on the unbound alpha rings. While the authors did not specify allosteric changes related to the catalytically active beta sites, their in vitro activity assay revealed the activation of the CT-L activity by approximately 3–4-fold. However, the other activities of the proteasome were not tested. In both structures, PA200 sits directly on the α-rings of 20S, closing the direct access to the 20S but partially opening the 20S entrance pore.

The high resolution of the PA200/20S complexes revealed two positively charged grooves on the exterior of PA200 that formed potential substrate entry sites (Figure 2B). However, these channels were obstructed by two negatively charged densities: 5,6[PP]2-InsP4 ((5,6)-bisdiphosphoinositol tetrakisphosphate); and InsP6 (Inositol hexakisphosphate) (Figure 2C,D, respectively). The substrate entry via PA200 might thus be fine-tuned by these highly negatively charged small signaling molecules. In the previous structures, InsP6 was involved in the structure stabilization, ternary interactions, and folding [25,26]. InsP6 was also reported to play a role in the RNA editing [25], mRNA transcription [27], RNA export [28], and DNA repair [29,30] and the regulation of the histone deacetylases (HDAC) activity [31]. InsP6 also acts as a glue by bringing Cullin-RING ligase (CRL) and COP9 signalosome (CSN) together, and plays a role in UV radiation resistance [32]. This regulatory function of InsP6 on multiple nuclear pathways accords with its high concentration in the nucleus [33]. Not much is known about the function of InsP4. Unfortunately, there is currently no PA200 structure without the presence of these molecules. Therefore, one can only surmise on their functions. Considering that PA200 is also located in the nucleus, InsP6 might interact with PA200 and fine-tune its function, potentially acting as an inhibitor. The above structural data thus provide insights that the PA200 binding might: (1) block large and positively charged substrates from entering; (2) increase selectivity towards the negatively charged substrates; and (3) potentially increase the catalytic activity for the ubiquitinylated substrates in hybrid complexes with 26S.

## 4. Function

The above-mentioned structural data provide evidence for an opening of the 20S alpha subunit gate and the rearrangement of the proteolytic sites upon the binding of PA200 to the 20S, which facilitates the proteolytic processing of the peptide substrates [4]. Which of the active sites are activated by PA200, however, is controversial. The early biochemical evidence, using PA200 and 20S isolated from bovine testes [1], indicated that PA200 activates the peptide hydrolysis of all of the 20S active sites, but predominantly the C-L activity. The group of Naveen Bangia also reported elevated C-L activity related to the PA200 expression in cells [21]. In stark contrast, the structural data summarized above demonstrate the activation of the T-L active site in an in vitro reconstituted PA200/20S complex but the reduction of the C-L and CT-L activities [4]. Guan et al., however, reported three–four-fold activation of the CT-L activity using recombinant human PA200 and human 20S isolated from red blood cells [9]. A recent BioRxiv study reported the activation of the C-L and inhibition of the T-L activities in an in vitro assay using cell extracts and supplemented recombinant human PA200 [12]. To further add to the confusion, a recent paper reported an increased activity for all of the catalytic subunits of the 20S when the recombinant human PA200 was added [34]. As the opening of the PA200 dome is small, the entry of the substrates via the PA200 gate would only be allowed for unstructured protein chains or peptides. Even more intriguing, the entry channels into the PA200 appear to be obstructed by highly negatively charged inositol phosphates [4,9]. The difference in the data raises the critical question of which of the substrates are degraded by the complexes containing one or two of the PA200 complexes attached to the 20S core. Moreover, what is the function of PA200 in a 26S hybrid proteasome complex containing the 20S bound to PA200 on one side and the 19S regulator on the other? In that case, the protein substrates would then most probably enter via the 19S regulator into the 20S core for degradation [35]. Does PA200 then act as a "flusher, aiding exit of peptide products through a widened orifice", as suggested by Michael Glickman, and similar to the PA28 proteasome activator family [3]?

One of the few concordant observations on the PA200 function relates to its role in sperm cell differentiation. Khor et al. found that the ubiquitous genetic depletion of PA200 in mice causes male infertility [7]. This observation was confirmed in an independent PA200 KO model [10]. Due to the predominant expression of PA200 in testes, it was suggested that PA200 preferentially associates with the spermatoproteasome, a specialized type of proteasome in which a gamete-specific α4s subunit replaces the α4 isoform of the constitutive proteasomes. This hypothesis was tested in the Bousquet lab: PA200 was enriched two-fold in the spermatoproteasomes (s20S) compared to the constitutive proteasome (c20S) when co-immunoprecipitated with an α4s antibody from bovine testes. However, the s20S and c20S levels were comparable when co-immunoprecipitation was completed with the PA200 antibody, suggesting that PA200 does not preferentially associate with one 20S type. The authors concluded that, although PA200 seemed to have a function in s20S, it does not act as an exclusive activator during the germ-cell differentiation, as 19S binds more to α4s [34].

The PA200 binding to 20S was found to be increased at least 10-fold in the spermatids (SPTs) and Sertoli (SER) cells compared to spermatogonia (SPG), suggesting a specialized function for PA200 in these cells [34]. A function for PA200 in the acetylation-associated degradation of core histones in response to the DNA double-strand breaks was proposed by Qian et al. [10]. In their study, the testes of PA200 KO mice showed prominent defects in the removal of core histones at the early stage of elongated spermatids [10]. In the pulldown assays, the authors observed the in vitro binding of acetylated histones to the recombinantly expressed bromodomains from mouse PA200 and its yeast homolog Blm10. The acetylated histones were also degraded in vitro in the presence of bovine PA200 by 20S proteasomes. In accordance with this notion, Mandemaker et al. observed elevated levels of histones upon the silencing of PA200 in the UV-exposed HeLa cells [11]. Another recent paper also proposed that PA200 regulates the stability of the histone marking (H3K4me3 and H3K56ac) in aging and transcription [36]. These results are intriguing, but hampered by the recent cryo-EM structures of human recombinant PA200, which did not confirm the presence of a bromodomain in PA200. Moreover, the two positively charged PA200 entry pores might not favor the binding of acetylated histones due to the predominance of positively charged histidine residues [4,9,10]. A very recent study showed that PA200 degrades acetylated-YAP1 in the in the nucleus of mesenchymal stem cells (MSC) in response to the histone deacetylase (HDAC) inhibitor apicidin [37]. The authors also injected PA200 KO MSCs into an infarcted heart to define the role of PA200 in myocardial infarction. Their results showed that, to maintain the therapeutic function of MSCs in myocardial infarction, the acute degradation of YAP1 in the nucleus by PA200 is necessary. The study by Douida et al. also suggested that the function of PA200 is probably not restricted to acetylated histone degradation [15]. They observed the binding of PA200 to genomic DNA that only partially overlapped with the presence of H3K27ac marks in a ChIP-seq analysis of a neuroblastoma cell line. Accordingly, other studies have implicated PA200 in DNA damage repair [14,21], mitochondrial stress responses [15], responses to proteasome inhibition [19], the glutamine sensitivity of cancer cells [38], myofibroblast differentiation [5], and, potentially, in the dampening of the MHC class I antigen presentation in lung cancer [12]. Most of these studies used the acute silencing of PA200 to investigate the potential functional effects of PA200 depletion in different cell types. As also discussed below in detail, the silencing of PA200 is generally well tolerated by the cells at baseline, but appears to be critical in response to stress. The knockdown of PA200 reduced the survival of cancer cells upon exposure to ionizing radiation and was associated with increased genomic instability and increased sensitivity towards glutamine depletion [14,21,38]. However, the embryonic stem cells isolated from PA200 knockout mice showed no increased sensitivity upon genotoxic stress (radiation of bleomycin), or altered mortality when crossed to p53-deficient mice [7]. In several rat models, the downregulation of PA200 in the endothelial cells by miRNA-29b was associated with increased oxidative stress and endothelial dysfunction [39]. This effect was, however, not unambiguously ascribed to the regulation of PA200, but might be due to additional and/or alternative targets of miRNA-29b. The stable silencing of PA200 in the neuroblastoma cells resulted in a metabolic shift from oxidative phosphorylation to glycolysis and elevated levels of intracellular ROS [40]. The regulation of mitochondrial function by PA200 was also supported by the ChIP-seq data of the same neuroblastoma cell line that demonstrated the binding of PA200 to the promotors of genes involved in cell-cycle progression and apoptosis in response to mitochondrial stress [15]. The silencing of the PA200 then sensitized cells to the rotenone-induced cellular death [15]. In contrast, the silencing of PA200 in primary human lung fibroblasts protected them from staurosporine-induced apoptosis and promoted myofibroblast differentiation [5]. The PA200-depleted mice did not show an altered response to bleomycin-induced lung fibrosis [5]. Taken together, the available literature on the possible cellular function of PA200 does not paint a consistent picture of the role of this proteasome activator in cell biology.

## 5. Dysregulation in Disease

Recent evidence unraveled a dysregulation of PA200 in disease. A database analysis for the presence of genomic mutations in PA200/PSME4 in cancer [41] revealed multiple alterations throughout the entire gene (Figure 3A), including mutations and amplifications in many different cancer types (Figure 3B). The functional role of these mutations or amplifications in carcinogenesis, however, remains unresolved.

Intriguingly, the PA200/PSME4 gene expression consistently increases across many different tumors (Table 1). This adds to the potential role of the PA200/proteasome complexes in oncogenesis. Given that most of the proteasomes in proliferating cells are localized in the nucleus [44], PA200 may support cell proliferation by facilitating the degradation of the specific nuclear substrates. This is consistent with the role of PA200-containing proteasomes in the rapid breakdown of the acetylated core histones, as mentioned earlier [10,11], which is required for the resolution of the DNA replication stress frequently occurring in cancer cells and caused by aberrant replication forks, stalled replication forks, or both [45]. PA200 has also been recently reported to activate mTORC1 [46], an anabolic signaling pathway that is constitutively triggered in many different types of cancers [47]. In this study, Ge and colleagues show that PA200/PSME4 gene silencing in hepatocellular carcinoma results in reduced mTOR phosphorylation and decreased cell proliferation in vitro. The molecular mechanisms by which PA200 promotes the mTORC1 activation remain obscure, but it is conceivable that this process occurs through an increased supply of free amino acids. Indeed, the proteasome-mediated protein degradation represents a significant source of peptides that are further degraded into amino acids by various intracellular peptidases [48]. As such, given the ability of PA200 to increase the proteasome activity [21], one could assume that tumor cells overexpressing PA200 produce high levels of free amino acids, which in turn activate mTORC1 following sensing by SESN2 and CASTOR [49]. This assumption would imply that the mTORC1 signaling is upregulated by increased proteasome function and/or protein turnover, rather than PA200 itself. Similar observations should be completed in the cells and tissues overexpressing PA28α/β, immunoproteasomes, or both, although this remains to be formally demonstrated.

In contrast to cancer, the neurodegenerative diseases were associated with low expression levels of PA200 (Table 1) [50,51,52]. This observation might relate to the global downregulation of the UPS function that typically accompanies these disorders [68]. However, the precise contribution of a reduced PA200 expression to the pathogenesis of neurodegeneration remains ill-defined. Given the ability of PA200 to increase proteasome activity by forming hybrid complexes with the 20S and 19S particles [21], it is tempting to speculate that PA200 actively participates in the breakdown of the ubiquitin-modified substrates. This assumption also precludes that PA200 depletion could actively contribute to the formation of the ubiquitin-positive inclusions inherent to neurodegenerative diseases. Depending on the type of neurodegeneration, these insoluble structures may aggregate in various subcellular compartments, including the nucleus [69], mitochondria [70], and cytoplasm [71], albeit the latter represents the predominant localization of these depositions. The observation that PA200 is mainly localized in the nucleus [1,72] suggests that the PA200/proteasomes are not primarily involved in the removal of cytosolic proteins. This also presupposes that the PA200 deficiencies in neurodegenerative diseases would only be marginally involved in the formation of the ubiquitin deposits. Alternatively, it is also conceivable that the PA200 dysfunction might contribute to neurodegeneration via its function in DNA repair [1]. Indeed, thanks to their potential ability to clear acetylated histones which arise in response to DNA damage, the PA200/proteasome complexes might contribute to maintaining genomic stability, even under challenging conditions [21,73]. As such, the cells devoid of PA200 may be particularly prone to DNA injuries and the subsequent structural changes that may affect the integrity of their genomes. These perturbations are likely to result in the uncontrolled leakage of nuclear DNA fragments into the cytosol, a process that triggers auto-inflammation following the sensing of the self-nucleic acids by cytosolic pattern recognition receptors (PRR) [74,75]. Consequently, the PA200 downregulation may contribute to the disease pathogenesis by facilitating neuroinflammation, whose continuous presence is understood to be a significant driver of neurodegeneration [76]. In support of this notion, it was recently shown that the PA200 gene silencing in murine lung adenocarcinoma was associated with an increased expression of proinflammatory cytokines, including type I and II interferon (IFN), TNF-α, and IL-17 [12]. However, the molecular mechanisms downregulating PA200 during neurodegeneration remain to be fully determined.

As shown in Table 1, our literature survey on PA200 revealed that its expression might be repressed during viral and/or bacterial infections. Herein, Minor and colleagues showed that the hepatitis B virus (HBV)-derived HBx protein increases the PA200 breakdown following activation of the cullin 4 DDB1 E3 ubiquitin ligase complex (CRL4) [53]. In this study, the authors suggest that HBV may benefit from PA200 depletion in host cells by preserving the pool of acetylated histones, which may favor the transcription of the circular viral DNA [53]. Likewise, PA200 is downregulated in neutrophils upon ingestion of *Staphylococcus aureus* [55]. Although the underlying molecular mechanisms remain unclear, it is conceivable that the suppression of PA200 by *S. aureus* represents an evasion strategy contributing to neutrophil destruction. This notion is supported by the view that the PA200/20S proteasome complexes are associated with increased oxidative stress [39,77]. Hence, it is seductively easy to imagine that the neutrophils devoid of PA200 following *S. aureus* uptake are less capable of generating the oxidative stress conditions required for efficient bacterial killing. While these two studies point to a potential role of PA200 in innate immunity, its contribution to the initiation of adaptive immune responses is much less clear. Notably, the role of the PA200-containing proteasomes in supplying viral and/or bacterial MHC class I-restricted peptides remains obscure. The lack of investigations in this regard is even more intriguing, considering the PA200 promotor contains STAT1- and STAT3-binding sites (Meiners, unpublished data). This observation indeed suggests that PA200 may be regulated in the cells via autocrine and/or paracrine type I and/or II interferons (IFN) loops, including professional antigen-presenting cells (APC), such as dendritic cells (DC), before T-cell priming. However, the early studies have refuted this assumption by showing that the transcription rate of PSME4 remained unchanged in DC following exposure to various maturation-inducing agents, including LPS, Poly-IC, CD40L, a combination of TNF-α, IL-1β, IL-6, and PGE2 [78]. This notion is further supported by recent work showing that IFN-γ fails to upregulate PA200 in colorectal cancer patient-derived organoids [79]. The reason for the lack of responsiveness of PA200/PSME4 to IFN signaling and/or microbial stimuli is unclear, but may be related to epigenetic processes. While these data do not exclude a potential role for the PA200-containing proteasomes in MHC class I antigen processing, they strongly suggest that these complexes do not shape the MHC class I peptide repertoire during DC maturation. Interestingly, a recent study even raised the possibility that the PA200/proteasome complexes might negatively affect the MHC class I antigen presentation [12]. In this work by Javitt et al., the overexpression of PA200 in the A549 lung tumor cell line was associated with a reduction in the MHC class I-restricted peptides, suggesting that PA200 might destroy tumor epitopes by modulating the three proteasome catalytic activities.

As shown in Table 1, the cells exposed to hypoxia were found to reduce their PA200 expression levels [56,58]. Since low oxygen levels typically trigger oxidative stress [80], the downregulation of PA200 under these conditions may be part of the antioxidant response. One could envision that the hypoxic cells decrease PA200 to support the assembly of alternative proteasome complexes, including the PA28α/β- and PA28γ-associated immunoproteasomes, which are more efficient at coping with the oxidant-damaged proteins [77,81]. This notion is in line with the fact that hypoxia also induces the inducible immunoproteasome subunits [82]. Intriguingly, the view that PA200 and the inducible subunits β1i, β2i, and β5i are transcriptionally differentially regulated suggests that PA200 may not preferentially associate with the proteasomes carrying immunoproteasomes subunits. However, this assumption remains to be fully demonstrated. Not surprisingly, decreased PA200 transcripts were found in the blood during acute ethanol exposure (Table 1) [57]. However, given the overall negative impact of ethanol on proteasome function described in the macrophages [83,84], it is likely that this effect is not specific to PA200 but applies to all of the proteasome subunits and activators.

## 6. Conclusions

Although some progress has been achieved in recent years, our current understanding of PA200/PSME4 biology is still in its infancy. Our knowledge about its function is indeed much less advanced than that of the other proteasome activators PA28α/β and PA28γ. One reason for the particularly small amount of available data on PA200/PSME4 today is likely due to the lack of suitable tools in previous years—notably specific antibodies—typically required for the design of reliable and functional investigations. However, the transcriptional studies have revealed specific gene expression patterns under certain circumstances, with PA200/PSME4 being consistently induced in cancer and repressed during infection. While the upregulation of PA200/PSME4 in tumor cells can be easily explained by its described function in DNA repair, its reduced expression levels in response to pathogens and/or inflammatory stimuli are enigmatic. This observation is even more intriguing, considering the fact that the expression levels of other proteasome activators (i.e., PA28α/β), on the contrary, are increased under these conditions. Our analysis, therefore, raises the possibility that the proteasomes may exert opposite functions depending on the capping activator, particularly concerning the MHC class I antigen processing. In this regard, further studies are warranted to address this hypothesis and determine the relevance of PA200/PSME4 as a potential therapeutic target in cancer and autoimmune diseases. The targeting of the PA200-bound 20S complexes offers the intriguing possibility of increasing specificity and reducing toxicity compared to broad proteasome inhibition. The latest advances in the structure of PA200/PSME4, both alone and in complex with the 20S, provide a strong structural biology rationale for the design of inhibitory molecules. The structure of PA200 revealed two putative entrance channels. Therefore, designing small molecules to block these entries could be a potential strategy. However, the grooves of these two openings are highly positively charged and blocked by two negatively charged inositol phosphates. Considering that these two molecules were observed in two independent cryo-EM structures [4,9], it is highly likely that they are present in the cell and have a regulatory function. Any small molecule targeting these grooves thus needs to be carefully designed for higher binding efficiency to replace the inositol phosphates. Another strategy to target the PA200/20S complexes involves blocking the contact sites between PA200 and the 20S. The two anchor points of PA200, one in the α5-α6 interface and one in the α1-α2 interface, could be targeted by small molecules to decrease the binding efficiency of PA200. However, as other proteasome activators also use this interface to bind to the 20S catalytic core, such inhibitory molecules might lack specificity.

Finally, one might consider that PA200/PSME4 has additional cellular functions independent of its binding to the proteasome. The evidence for free PA200 in the cell is scarce and indirect. Welk et al. observed the fast recruitment of PA200 to 20S upon Bortezomib treatment without any increase in the protein or RNA levels [19]. This finding might thus indicate the presence of free PA200 in the cell, which is rapidly recruited to the proteasome under stress conditions. This free PA200 might also exert proteasome-independent roles in the cell, which need to be further explored.

## Figures and Tables

**Figure 1 biomolecules-12-01150-f001:**
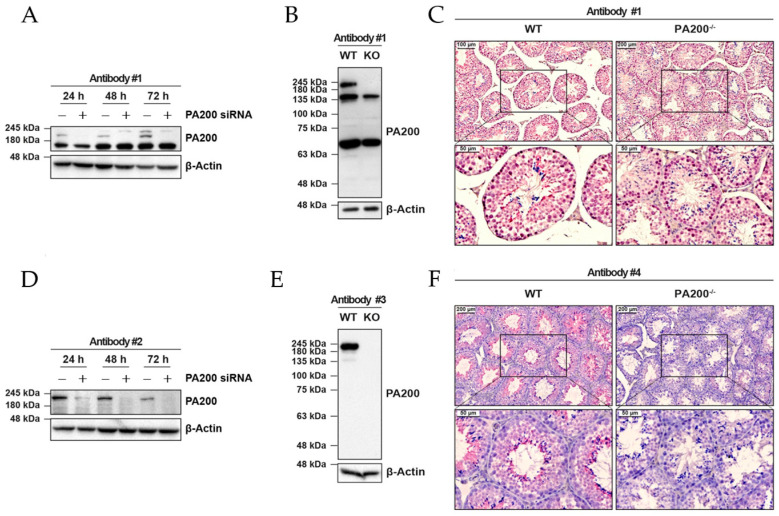
Analysis of the specificity of commercially available anti-PA200 antibodies. (**A**) A time course of transient PA200 silencing in human A549 alveolar epithelial cells was analyzed for specific recognition of PA200 by Western blotting using the leading commercially available anti-PA200 antibody #1 (PA1-1961, Thermo Fisher Scientific, Waltham, MA, USA); (**B**) Protein lysates of testes were prepared from wild type (WT) and PA200^-/-^ (KO) mice (obtained from [7]) and analyzed for detection of PA200 by Western blotting using antibody #1; (**C**) Immunohistochemistry (IHC) staining of paraffin-embedded testis sections from wild type (WT) and PA200^-/-^ (KO) mice with antibody #1; (**D**) Samples used in (**A**) were analyzed with the specific PA200-targeting antibody #2 (NBP1-22236, Novus Biologicals, Littleton, CO, USA); (**E**) Samples used in (**B**) were analyzed with the specific PA200-targeting antibody #3 (NBP2-32575, Novus Biologicals, Littleton, USA); (**F**) Tissue sections from wild type (WT) and PA200^-/-^ (KO) mice were stained with antibody #4 (sc-135512, Santa Cruz, Dallas, TX, USA) by IHC. Figures show representative results for experiments performed with *n* = 3. (Reproduced with permission from Welk et al., Scientific Reports; published by Nature Publishing Group, 2019 [5]).

**Figure 2 biomolecules-12-01150-f002:**
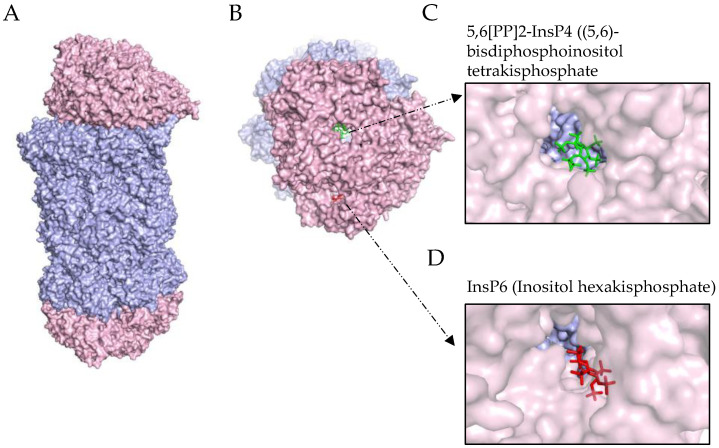
Structure of PA200. (**A**) Overall view of PA200-20S-PA200 complex structure (PA200 (pink) and 20S (blue) structure PDB ID: 6REY); (**B**) Top view of PA200 with two negatively charged molecules; (**C**) 5,6[PP]2-InsP4 ((5,6)-bisdiphosphoinositol tetrakisphosphate; and (**D**) InsP6 (Inositol hexakisphosphate).

**Figure 3 biomolecules-12-01150-f003:**
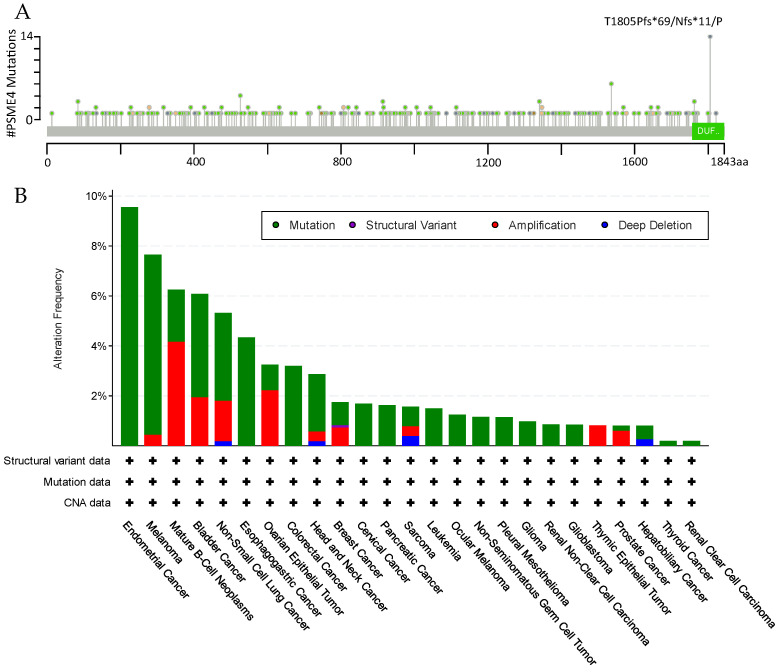
Mutations in the PA200/PSME4 gene. (**A**) Lollipop mutation diagram, using mutation data from the TCGA PanCancer Atlas Studies. The matching mutation types are reflected in the coloring of the circles on the mutation diagram. The circle’s color is chosen according to the mutation type that occurs most frequently when multiple mutation types occur at the same site. Missense mutations (light green, unknown significance), Truncating mutations (light grey, unknown significance), Inframe mutations (dark brown, unknown significance), Splice mutations (light brown, unknown significance), SV/Fusion mutation (light pink); (**B**) PA200/PSME4 alteration frequencies in different cancer types as obtained from the TCGA PanCancer Atlas Studies (data and figures retrieved from cBioPortal on the 16th of August, 2022 [41,42,43]).

**Table 1 biomolecules-12-01150-t001:** An overview of PA200/PSME4 gene regulation in health and disease. (↓ indicates downregulation, ↑ indicates upregulation).

Context	Disease	PA200/PSME4 Expression	Reference
Neurodevelopment and Neurodegeneration	Parkinson’s disease	↓	[50]
	Pick disease	↓	[51]
	Ataxia-Telangiectasia	↓	[52]
Infection	Hepatitis B virus infection	↓	[53]
	Herpes simplex virus G207 infection	↓	[54]
	Staphylococcus aureus infection	↓	[55]
Environmental stresses	Hypoxia	↓	[56]
	Acute ethanol exposure	↓	[57]
	Lactic acidosis and hypoxia	↓	[58]
Cardiovascular diseases	Endothelial dysfunction	↑	[39]
Cancer	Multiple myeloma	↑ (loss of miR-29b)	[20]
		↑	[59]
	Gastric cancer	↑	[60]
	Esophageal squamous cell carcinoma	↑	[61]
	Esophageal Adenocarcinoma	↑	[62]
	Oral squamous cell cancer (OSCC)	↑	[63]
	Hepatocellular carcinoma	↑	[46]
	Non-small lung cancer	↑	[64]
	Lung cancer	↑	[65]
	Transitional cell carcinoma of the kidney	↑	[66]
	Osteosarcoma	↓	[67]

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
