# Peer review of "The Proteasome Activator PA200/PSME4: An Emerging New Player in Health and Disease"

_biomolecules, 2022, doi:10.3390/biom12081150_

Round 1

Reviewer 1 Report

This is a comprehensive, timely and well-organized review on the function of PA200. The authors’ efforts in clarifying some of the technical issues in PA200 research are particularly welcome and appreciated. It’d be nice if the authors could include some minor additions/corrections to the current manuscript, as detailed below:

1.     The discussion was entirely focused on the role of PA200 in proteasome regulation, whereas proteasome-independent functions of PA200 are possible.

2.     In addition to deletion or upregulation of the protein as a whole, other mechanisms for PA200 regulation such as point mutations and post-translational modifications have been documented in relevant databases.

Although the picture is much less clear in the above regards, some brief discussions along these lines would provoke new thoughts and may attract the attention from researchers outside the proteasome field.

3.     Some typos: Line 40 “it”; Line 111 “Hela”.

Author Response

Reviewer 1:

  1. This is a comprehensive, timely and well-organized review on the function of PA200. The authors’ efforts in clarifying some of the technical issues in PA200 research are particularly welcome and appreciated. It’d be nice if the authors could include some minor additions/corrections to the current manuscript, as detailed below:

R1: We are delighted about this positive comment by the reviewer and have modified the manuscript according to the reviewer’s suggestion. All changes are marked in track change modus.

  1. The discussion was entirely focused on the role of PA200 in proteasome regulation, whereas proteasome-independent functions of PA200 are possible.

R2: We thank the reviewer for this important comment. Indeed, we here focused on the available data on the role of PA200 as a proteasome activator. Evidence for the existence of free PA200 is scarce and rather indirect. We have therefore added some sentences on its potential proteasome-independent role in the outlook.

  1. In addition to deletion or upregulation of the protein as a whole, other mechanisms for PA200 regulation such as point mutations and post-translational modifications have been documented in relevant databases. Although the picture is much less clear in the above regards, some brief discussions along these lines would provoke new thoughts and may attract the attention from researchers outside the proteasome field.

R3: This is indeed a very interesting thought and observation. According to the reviewer’s comment, we have now included a short section on PA200 mutations in cancer and added a figure that summarizes the available data on cbioportal (Figure 3).

  1. Some typos: Line 40 “it”; Line 111 “Hela”.

R4: We have corrected the typos and have proof-read the entire manuscript again.

Reviewer 2 Report

The authors did a good job of reviewing recent work on PA200's structure, biology, and function as well as how its malfunction is associated with diseases and discussing about prospective therapy options. It is important to note that the assessment of antibody specificity will be very helpful for future research on PA200. Given the biological importance of PA200 and the scarcity of reviews on this protein, this paper deserves to be published; nonetheless, the outlook in the conclusion section might have included additional comments, such as the proteasome-independent functions of PA200, and the potential use of the high resolution PA200 structure in the creation of chemical probes for scientific study and drug development.

Author Response

We thank the reviewer for his/her positive evaluation. Accordingly, we have included a short paragraph on the potential structure-guided inhibitor design strategies and also on the potential proteasome-independent functions of PA200 in our outlook part. All changes in the manuscript are marker in track change modus.